# PD-L1’s Role in Preventing Alloreactive T Cell Responses Following Hematopoietic and Organ Transplant

**DOI:** 10.3390/cells12121609

**Published:** 2023-06-12

**Authors:** Shane Handelsman, Juliana Overbey, Kevin Chen, Justin Lee, Delour Haj, Yong Li

**Affiliations:** BioMedical Engineering, Department of Orthopaedic Surgery, Homer Stryker MD School of Medicine (WMed), Western Michigan University, Kalamazoo, MI 49007, USA; shane.handelsman@wmed.edu (S.H.); juliana.overbey@wmed.edu (J.O.); kevin.chen@wmed.edu (K.C.); justin.lee@wmed.edu (J.L.); delour.haj@wmed.edu (D.H.)

**Keywords:** alloreactive, graft rejection, PD-L1, graft versus host disease (GVHD), autoimmunity

## Abstract

Over the past decade, Programmed Death-Ligand 1 (PD-L1) has emerged as a prominent target for cancer immunotherapies. However, its potential as an immunosuppressive therapy has been limited. In this review, we present the immunological basis of graft rejection and graft-versus-host disease (GVHD), followed by a summary of biologically relevant molecular interactions of both PD-L1 and Programmed Cell Death Protein 1 (PD-1). Finally, we present a translational perspective on how PD-L1 can interrupt alloreactive-driven processes to increase immune tolerance. Unlike most current therapies that block PD-L1 and/or its interaction with PD-1, this review focuses on how upregulation or reversed sequestration of this ligand may reduce autoimmunity, ameliorate GVHD, and enhance graft survival following organ transplant.

## 1. Introduction

Self-tolerance is developed by T cells in the thymus. Thymocytes with T cell receptors (TCR) that have a strong affinity for self-peptides presented by the major histocompatibility complex (MHC), or the MHC complex itself, undergo negative selection, resulting in self-reactive thymocytes that undergo apoptosis and do not mature into T cells. However, in organ transplant cases where the tissue within a host is a foreign entity, alloreactivity can develop as central tolerance does not develop towards the non-self/allogenic peptides [1,2]. This occurs in the case of chronic graft rejection, as portions of donor HLA molecules are presented by antigen-presenting cells (APC) in the same manner as foreign bacterial antigens. In acute rejection, TCRs instead identify whole human leukocyte antigen (HLA) molecules presented by the donor cells as foreign. Due to these rejection processes, almost all organ transplants have limited life spans, requiring immunosuppressive therapy in recipients to prolong their graft’s viability [3,4].

HLA matching is crucial for successful hematopoietic stem cell transplantation and to prevent graft-versus-host disease (GVHD). When there is a mismatch between the donor and recipient HLA, donor T cells may mistake host self-peptides for infectious agents due to a process called molecular mimicry. This happens when a donor MHC class I or II molecule displays a common self-peptide that is not normally immunogenic but appears infectious due to variances in the host’s MHC structure. In contrast, HLA-matched donors present alloantigens or minor histocompatibility antigens (mHAs), which can cause GVHD when they are presented by host MHC. mHAs are polymorphic peptides that differ between donor and host cells, and they are positioned within the peptide-binding groove on host MHCs. The binding of mHAs with donor TCRs can lead to an alloreaction where donor T cells attack host tissues [5,6]. GVHD affects up to 80% of mismatched hematopoietic cell transplant (HCT) recipients, and even fully HLA-matched related donors can still result in GVHD in approximately 40% of recipients [7,8].

In both HCT and organ transplantation, alloreactive T cells are activated, often targeting the foreign organ or organs through their presentation of alloantigen [9]. Therefore, a major challenge in these procedures is preserving functional host/graft tissues from alloreactive T cells. This immune response is primarily coordinated by the recognition of MHC and its associated antigen by APCs, which induce subsequent T cell responses. Depending on the subsequent presentation of co-stimulatory and/or co-inhibitory molecules by the APC in association with MHC, co-modulatory molecules can either enhance or weaken an inflammatory response against a target antigen [10]. This article will focus on how activation of T cells by the co-inhibitory molecule PD-L1 has been shown to reduce autoimmunity, symptoms of GVHD, and preserve organ transplants. It is worth noting that PD-L2, which has similar effects as PD-L1, has recently been discovered but will not be discussed here, as PD-L1 appears to be more biologically relevant and has been more widely studied.

## 2. Current Challenges in Transplantation

### 2.1. Graft versus Host Disease

One of the most common complications of hematopoietic transplantation therapy is GVHD. This is characterized by transplanted immune cells’ recognition of their new host as foreign and is followed by an immune response against the host following a HCT. GVHD is scored to describe the disease’s severity and functional impact on specific organs. This alloreactive condition can be categorized as either acute or chronic [11,12]

Acute graft versus host disease (aGVHD) is a major cause of morbidity and mortality in HCT. Up to 40% of all cell transplant recipients develop aGVHD [13]. Clinical manifestations of aGVHD most commonly occur within 100 days of transplantation, but acute cases can occur past this time point. Stage 1 aGVHD has limited skin involvement and can present with maculopapular rash, erythema, and pruritus. More severe diseases, stages 2–4, tend to involve the liver and/or gastrointestinal tract. This can present as hyperbilirubinemia, jaundice, and cholestatic hepatitis with liver involvement, as well as severe nausea, vomiting, weight loss, and diarrhea in the gastrointestinal tract [11,12,13,14,15]. However, extensive skin involvement is also possible, presenting as generalized erythroderma that covers greater than 50% of the body’s surface area with additional areas of bullous or desquamation [16].

The pathophysiological mechanism of aGVHD first involves generalized tissue damage induced by radiation or chemotherapeutic agents used as a conditioning regimen before stem cell transplantation [16]. This damage to host tissues releases Damage Associated Molecular Patterns (DAMPs), stimulating the release of a variety of inflammatory cytokines such as IL-1, IL-6, and TNF-α from host non-hematopoietic and remaining host hematopoietic cells. These cytokines upregulate MHC presentation and cell surface adhesion molecules on host APCs, priming the host’s organs for targeting by a subset of T cells that are termed “alloreactive” [17,18]. Focusing on the GI system, destruction of rapidly replicating epithelial cells in the intestine results in increased permeability of the mucosa, followed by the release of microbial components such as LPS and other pathogen-associated molecular patterns (PAMPs) into systemic circulation. Host DCs are then activated by these PAMPs and the aforementioned DAMPs. Recognition of these DAMPs and PAMPs by toll-like receptors causes these DCs to begin increasing the expression of alloantigen and CCR7, which allow for their migration into mesenteric lymph nodes. The upregulation of MHC, surface adhesion molecules, and migration of alloreactive DCs primes the transplant recipient for the initial onset of aGVHD [19,20,21].

Host and new donor APCs then activate alloreactive donor T cells still present in the hematopoietic transplant that aggregate in secondary lymphoid tissue [22,23]. This results in increased proliferation, trafficking, and activation of both CD4+ and CD8+ T cells, with T cell activation further inducing the release of IL-2 and IFN-γ [24,25,26,27,28]. Finally, alloreactive effector T cells (including helper T cells and cytotoxic T cells), along with natural killer (NK) cells, migrate to various organ systems where they begin to cause cytotoxic tissue damage. This eventually leads to the characteristic manifestations of aGVHD [29,30,31].

In contrast to aGVHD, chronic graft versus host disease (cGVHD) typically occurs 100 or more days after transplantation, affecting 30–70% of transplant recipients [32,33,34]. Clinical manifestations of chronic GVHD can appear in a variety of organ systems and can either be restricted to a single organ or widespread throughout the body. Dermatologic manifestations of chronic GVHD include lichen-sclerosis-like lesions, depigmentation, sweat impairment, and heat intolerance, along with the typical erythema and maculopapular rash. Gastrointestinal features of chronic GVHD include exocrine pancreatic insufficiency, webs, strictures, and stenosis within the esophagus. Pulmonary manifestations include bronchiolitis obliterans and bronchiectasis. Patients with chronic GVHD also experience dry and painful eyes, conjunctivitis, and punctate keratitis [11,12].

The pathophysiological mechanism of cGVHD has been suggested to follow a similar sequence, which culminates in the development of aGVHD [35]. Initially, host tissues are damaged by the transplant conditioning regimen, infectious agents, and previous aGVHD. This damage causes the release of various cytokines and microbial contents, which activate APCs [20,36,37,38,39]. The activation of professional APCs, such as macrophages and DCs, then leads to the subsequent differentiation of donor CD4+ T cells into Th1 and Th17 cells, which are then recruited into host tissues [40,41,42,43]. These effector T cells then infiltrate and cause widespread dysregulated inflammation in various organs, which further feeds into a cycle of inflammation and cell damage. In particular, the thymus and other secondary lymphoid organs are damaged, resulting in the loss of central and peripheral tolerance and the production of autoreactive immune cells [44,45,46,47,48,49,50].

This widespread tissue damage results in a cycle of perpetual inflammation [51,52,53,54,55,56,57,58,59,60,61,62,63,64]. Following the release of cytokines from damaged host tissues, activated immune cells, including APCs, T cells, B cells, and NK cells, continue damaging their host, which in turn perpetuates the release of cytokines. Regulatory T cells (Tregs) that normally act as a break to this inflammatory cycle are nonfunctional following the loss of central and peripheral tolerance [51,52,53,54,55,56,57]. Finally, once tissues have been eviscerated by alloreactive immune cells, there is abnormal tissue repair, which results in the release of TGF-β and widespread tissue fibrosis, ultimately manifesting in the symptoms of cGVHD [58,59,60,61,62,63,64].

aGVHD and cGVHD differ in their pathophysiology and symptomatology, with aGVHD characterized by cytotoxicity and cGHVD by tissue fibrosis. However, central to both processes is T cell activity, meaning an interruption to this T cell alloactivation would protect tissues from the effects of both aGVHD and cGVHD.

### 2.2. HCT Rejection

Graft failure is a severe complication following HCT that is divided into primary graft failure and secondary graft failure. In primary graft failure, there is a failure to achieve initial engraftment of the HCT. This is defined as an absolute neutrophil count (ANC) lower than 0.5 × 109/L by 28 days, platelet levels less than 20 × 109/L, or hemoglobin levels less than 80 g/L after transplantation. In secondary graft failure, there is a failure of the transplanted cells (ANC lower than 0.5 × 109/L) after initially successful engrafting [65].

The pathogenesis of graft failure in HCT is characterized by a complex, multifactorial, alloreactive immune response from host T cells and NK cells. Graft failure due to host NK cells most commonly occurs following MHC-mismatched HCT. However, even in HLA-matched HCT, host T cells have been shown to promote graft failure [66,67,68]. Host NK cells have inhibitory receptors that are unable to recognize the mismatched MHC class one molecules on the transplanted cells, resulting in NK-mediated cytotoxic rejection of the allogenic cells [69,70]. Host cytotoxic T cells remaining after the conditioning regimen in HCT also play a major role in graft failure. The exact mechanism behind host T cell-mediated graft rejection remains unclear, as the cytotoxic activity of these T cells is independent of typical perforin, Fas-ligand, TNF-related apoptosis-inducing ligand, and TNF Receptor-1 mechanisms [71,72,73,74].

Interestingly, donor NK cells and donor T cells have been shown to promote engraftment by inhibiting the host cell response to the transplanted cells [75,76,77,78]. Both host and donor Treg cells have also been demonstrated to facilitate engraftment in HCT through the production of IL-10 and TGF-β, suppressing T cell and NK cell-mediated graft failure [79,80,81].

Even though graft rejection has an extremely poor prognosis, the condition is relatively rare among allogeneic HCT recipients, with an incidence of roughly 5% [82]. The current treatment for HCT graft failure mainly consists of G-CSF therapy and additional allogenic cells to increase the graft’s proliferative capacity and prevent its rejection [83]. The rarity of HCT graft failure in comparison to rates of GVHD and solid organ rejection makes it no surprise that current research on how PD-L1 functions in HCT rejection is scarce itself.

### 2.3. The Pathophysiologic Process of Transplant Rejection

Graft failure remains a major limitation in other types of tissue transplantation beyond HCT. Kidney transplant is the most transplanted organ in the United States of America. In a review of United Network for Organ Sharing data on kidney transplantation from 2000 to 2014, 50,301 graft failures occurred, or about 3600 every year. Of this, 48% were due to chronic rejection, and 12% were due to acute rejection [84]. The immunological mechanism of transplant rejection occurs in three main forms: hyperacute, acute, and chronic. Hyperacute rejection (HAR) typically occurs within minutes to hours. Pre-formed circulating antibodies quickly bind to transplant tissue, typically resulting in irreversible graft thrombosis and ischemia. However, with the advent of screening technologies such as crossmatch testing and antibody screening over the last 40 years, HAR is now a rare occurrence [85]. Hyperacute rejection (HAR) is mainly a humoral immune response and is classified as a Type II hypersensitivity reaction. It occurs because pre-formed circulatory antibodies bind to the allogenic graft tissue. These antibodies arise because of previous exposure to foreign immune molecules, typically the constitutively expressed human leukocyte antigen Class I molecules. As such, HAR most frequently occurs in patients who have had previous transplantation, blood transfusion, or pregnancy [86].

In contrast, acute rejection (AR) occurs within days to weeks of transplantation. It is the result of the host immune system identifying the graft as foreign and destroying it, leading to graft failure. This type of rejection is largely controlled by immunosuppressive therapy, leading to a decreased incidence and better long-term outcomes. However, even with treatment, AR still occurs in about 7% of renal transplants every year [87]. Chronic rejection (CR) occurs months to years after transplantation and affects 100% of transplants to some degree. It is one of the main determinants of the longevity of solid organ transplants [88]. Its full etiology has yet to be understood, and proposed pathological mechanisms cite inflammatory, humoral, and cellular immune reactions.

Acute transplant rejection differs from HAR in that it involves both a humoral and cellular immune response. Additionally, in AR, the patient does not have pre-formed antibodies, but rather develops an activated immune response toward the graft tissue. This immune response arises because resident-donor antigen-presenting cells in the graft travel to sites of immune activation in the host, such as lymph nodes, and are recognized by a subset of T cells that are alloreactive. Alloreactive T cells are defined by the ability of their T cell receptors (TCRs) to bind non-self MHC molecules [89]. The mechanism of AR is the direct pathway of allorecognition, in that graft MHC molecules are directly presented to and recognized by host lymphocytes. Activation of alloreactive T cells leads to the typical host immune response to a foreign substance. This includes the activation of CD8+ T cells, Th1 T cells, Th2 T cells, and Follicular Helper T (TFH) cells. CD8+ T cells will migrate to the transplanted tissue and exert their direct cytotoxic effects. Activated Th1 cells will secrete IFN-γ resulting in the activation of macrophages and inflammation. TFH T cells serve as a link to humoral immunity, aiding B cell activation and the generation of donor-specific antibodies (DSAs) [90].

Whereas acute transplant rejection occurs via direct allorecognition, chronic transplant rejection occurs via the indirect pathway of allorecognition. The indirect pathway of allorecognition activates when graft T cells die in a manner that results in the release of cellular alloantigens, such as immune-mediated death or inflammasome-mediated pyroptosis. These cellular alloantigens, including peptide fragments of donor MHC molecules, are then phagocytosed by host APCs. Activated host immune cells travel to sites of immune activation and stimulate an immune response against the transplanted tissue, like the direct pathway of allorecognition [9]. While CR initially follows a similar cellular immune response activation as AR, there are several additional pathological changes that make CR a more complex and severe disease. One of the most distinctive features of chronic solid organ rejection is the activation of vascular smooth muscle cells and fibroblasts, resulting in intimal hyperplasia of blood vessels, and leading to ischemia of graft tissue. While the exact pathogenesis of this is not understood, most hypotheses start with vascular endothelial inflammation due to T cell-mediated release of cytokines. It has been suggested that this immune response can lead to aberrant migration and proliferation of vascular smooth muscle cells [88].

While the exact immunological mechanisms of AR and CR processes are varied, they both share an underlying framework: cellular damage to graft tissue that initiates an immune response driven by T cells. If these processes can be interrupted by limiting immune cell activation, it may promote longevity in engrafted tissues.

## 3. PD-L1 Mechanism of Action

PD-L1 (also known as B7-H1) and its corresponding receptor, PD-1, are both transmembrane proteins that are a part of the immunoglobulin superfamily. PD-L1 is constitutively present on the surface of hematopoietic cells such as monocytes and T cells. However, monocytes in peripheral blood do not tend to express PD-L1. Certain immune-privileged sites, such as the placenta and cornea, have also been shown to maintain expression of PD-L1. Additionally, the ligand’s expression is upregulated during active immune responses. This upregulation has been linked to the presence of pro-inflammatory cytokines such as IFN-γ or exogenous LPS on the surface of hematopoietic cells and at the surface of their epithelial and endothelial cells [28,91,92]. PD-L1 mRNA has been found in human organs, including the heart, kidney, and lung, but not in the colon or small intestine [93]. In contrast to this finding, immunohistochemistry stains performed in mice differ from the mRNA expression profiles seen in humans. Stains still show PD-L1 protein expression in the endothelium of the heart and macrophages in the lung. However, in contrast to the mRNA expression profiles of humans, mice express PD-L1 proteins in the lamina propria of the small intestine but not in the kidneys [92].

Although soluble versions of PD-L1 and PD-1 exist, conventionally, the ligand and its receptor are transmembrane-bound proteins that are displayed on the cell’s surface. More specifically, PD-1 has an extracellular domain that allows for interaction with PD-L1, a localizing transmembrane portion, and an intracellular domain that allows for signal transduction [94]. The receptor’s production is upregulated via TCR recognition of MHC, thereby preventing the overactivation of T cells and limiting immune-mediated damage to native tissue [95,96]. PD-L1 has been shown to decrease the expansion of CD4+ T cells and NK cells, limit the cytotoxic effects of CD8+ T cells, and simultaneously increase the differentiation of CD4+ T cells into Tregs [97,98,99]. PD-1 can be stimulated following TCR activation in a two-step sequence whereby TCR engagement with MHC is followed by co-inhibition by PD-L1 [100]. Once PD-L1 is recognized by PD-1 in peripheral (non-immune) tissues, it initiates a co-inhibitory signal, limiting T cell proliferation (Figure 1b). This immunomodulatory effect is driven by the downregulation of a variety of intracellular pathways responsible for cellular metabolism and proliferation.

Crucial to this intracellular regulation are intracellular signaling domains on the PD-1 molecule known as immunoreceptor tyrosine-based switch motifs (ITSM). Following TRC activation and PD-1/PD-L1 binding, these ITSMs are phosphorylated, allowing for their interaction with a cytoplasmic tyrosine phosphatase called Src homology 2 domain-containing phosphatase 2 (SHP-2). SHP-2 contains an enzymatic tail portion that allows for downstream inhibition of phosphatidylinositol-3-kinase (PI3K) and Akt as well as a reduction in IL-2 production [101,102,103,104]. PD-L1’s inhibition of PI3K/AKT thereby directly counteracts the effects exerted by CD80 (also known as B7-1), a costimulatory molecule present on APCs [105,106] (Figure 1a). This prevents T cell activation via two alternate mechanisms: the loss of IL-2, preventing T cell expansion, and the limitation of the PI3K/AkT cell-signaling cascade, reducing glucose metabolism. Thereby, CD8+ and CD4+ T cells are prevented from proliferating and exerting their effector functions. This cumulative effect allows PD-L1 to act as a break on lymphocyte activity, preventing an overactive immune response [107,108,109].

There is an additional interaction between PD-L1 and another B7 immunoregulatory ligand, CD80. This interaction has not been as extensively studied as the PD-L1/PD-1 pathway, so the mechanism of the PD-L1/CD80 interaction is not entirely clear. Reports on the PD-L1 binding of CD80 have differed, but recent data indicates that their interactions are restricted to *cis*-membrane interactions and that the molecules do not interact *trans*-cellularly. This means that CD80 and PD-L1 are restricted to interactions on the same cellular membrane as they combine to form a heterodimer [110,111].

The CD80/PD-L1 interaction on the surface of APCs is pro-inflammatory, as CD80 inhibits PD-L1’s binding of PD-1, while CD80’s capacity to bind CD28 is preserved. That is to say, in the presence of both equal portions of CD80 and PD-L1, CD80 costimulatory signaling would tend to predominate on the surface of professional APCs [110,111,112]. However, the PD-L1/CD80 interaction also seems to be involved with the promotion of Treg expansion and T cell tolerance [113,114]. Additionally, the expression of CD80 by CD8+ T cells prevents this population of cells from proliferating [115]. Taken together, these findings support the theory positing that the effect of the PD-L1/CD80 interaction is dependent on the cell type that expresses the two proteins. Although the interaction between PD-L1 and CD80 offers PD-L1 an additional pathway to regulate T cell survival, this is a relatively new discovery, and the exact mechanisms underlying these molecules’ interaction have yet to be elucidated [110]. 

## 4. PD-L1 Amelioration of GVHD, Autoimmunity, and Graft Rejection

### 4.1. GVHD

A major goal of therapeutics following hematopoietic transplantation is to prevent the development of GVHD. PD-L1 appears to have the potential to promote immune tolerance following transplant and minimize the potential for developing adverse reactions caused by stem cell therapy. Tang et al. demonstrated how in vivo hydrodynamic gene transfer, which induces overexpression of PD-L1 in murine aGVHD models, results in lower lethality due to aGVHD. This overexpression of PD-L1 was found to result in the inhibition of donor T cells and reduce effector memory status. Moreover, PD-L1 overexpression resulted in murine models exhibiting lower levels of pro-inflammatory cytokines secreted from effector T cells, less proliferation, and increases in the apoptotic activity of these effector T cells [116].

A similar experiment performed by Blazar et al. found that blocking the PD-1 receptor from engaging with PD-L1 accelerated the lethality of GVHD in murine models. Following GVHD induction using donor splenocytes, anti-murine PD-1 monoclonal antibodies (mAB) or PD-L1 fusion proteins were administered, which block PD-1 signaling with its ligand. The PD-1 blockade was found to enhance T cell alloresponsiveness both in vitro and in vivo. This alloresponse was associated with a significant increase in inflammatory cytokines such as TNF-α and IFN-γ and was thought to cause a significant acceleration in GVHD lethality. Notably, analysis of the murine liver tissue revealed that irrelevant mAB (meaning non-PD-1-blocking)-treated recipient mice had less severe injuries when compared to those given anti-PD-1 mAB [117].

It is important to note that the development of GVHD is dependent on which cells express PD-L1. While expression of the ligand on donor T cells induces GVHD, expression on parenchymal tissue such as hepatocytes results in protection from the development of GVHD [118,119].

Deng et al. further characterized the PD-L1/CD80 interaction by administering anti-PD-L1 monoclonal antibodies. These antibodies were designed to block PD-L1’s interaction with CD80 without interfering with the PD-L1/PD-1 pathway. Administration of the antibody was performed on mice with intact PD-1 and rat-IgG in a control group for five days following a HCT [114]. After inducing GVHD via injection of donor spleen cells, Deng et al. found that in mouse models with intact PD-1, administration of these blocking antibodies reduced proliferation and apoptosis of donor alloreactive T cells. Overall, this resulted in a relative reduction in proliferation that was less marked than the reduction in apoptosis and allowed for an expansion of alloreactive donor T cells, producing more severe GVHD. The experimental group showed 100% mortality by day 7, following the administration of the PD-L1 antibody, while 50% of the control group survived for >50 days (N = 10). Cellular analysis showed that there was a marked increase in CD4+ donor T cells in the experimental group, as well as a decrease in expression of PD-1, quantified by RT-PCR. Cytokine analysis following blockage of PD-L1/CD80 showed that there was a significant decrease in IL-2 production as well as an increase in IFN-γ and TNF-α found in the serum. The increase in IFN-γ and TNF-α was attributed to increased numbers of alloreactive T cell populations in the experimental cohort. Additionally, the decreased IL-2 concentration was associated with decreased PD-1 production by both CD4+ and CD8+ T cells [120]. Intracellularly, the block on PD-L1/CD80 interactions increased the expression of BCL-xL, a pro-survival gene, and reduced the expression of caspase-3 in donor T cells. Notably, the ameliorative effect that the PD-L1/CD80 interaction has on GVHD depends on the presence of PD-1 interactions. Blocking the PD-L1/CD80 interaction in PD-1^−/−^ mice instead alleviates GVHD when compared to PD-1^−/−^ mice that had not received the blocking antibody. Overall, the results from this experiment demonstrated that the PD-L1/CD80 interaction prevents GVHD in a PD-1-dependent manner [111].

Deng et al. also found that injection with PD-L1 fused to an Ig Fc domain led to an improvement in GVHD symptoms in mice with intact PD-1. This is because there is a crucial balance between proliferation and apoptosis in alloreactive T cells. In a PD-L1^−/−^ cohort of mice, it was found that following injection of a PD-L1 Ig-expressing plasmid, there was a significant reduction of donor CD4+ T cells in the spleen and liver of the experimental group, resulting in less severe GVHD. In comparison, the control group without the PD-L1 plasmid injection developed severe acute GVHD and died within 7 days of the transplant. Interestingly, this experiment was repeated with a PD-1^−/−^ and control group model that contrasted these results. In PD-1^−/−^ mice, injection of a PD-L1 plasmid led to rapid loss of body weight and death 15 days post-HCT. Overall, there was an increase in the severity of GVHD, as demonstrated by the increase in mortality, symptomatology, and expansion of alloreactive donor CD4+ T cells. The results of Deng et al. reinforce the notion that the interaction between PD-L1 and CD80 leads to augmented proliferation of activated T cells and that PD-L1’s interaction with the PD-1 receptor causes a crucial increase in alloreactive T cell apoptosis. This balance is paramount to survival post-transplant [111].

### 4.2. Autoimmunity

The mechanism by which PD-L1 plays a role in autoimmune disease can be translated thematically to the expression of PD-L1 during cell transplantation. A study by Hu et al., using DBA/1J mouse models (a strain of mice bred to develop rheumatoid arthritis), studied the impact of PD-L1 overexpression on these mice. Researchers administered a PD-L1 lentiviral vector to transfect mouse bone marrow mesenchymal cells, which induced PD-L1 overexpression in the cells. DBA/1J mice that were treated with mesenchymal stem cells that had received the PD-L1 vector exhibited less joint damage, less activation of pro-inflammatory cytokines, and an inhibition of T and B cell activation. Moreover, there was limited cartilage damage, synovial hyperplasia, inflammatory cell infiltration, and bone erosion in the mice treated with the PD-L1 overexpression vector. These transplanted stem cells decreased the DCs found in mouse joints and increased the Tregs in the tissue, promoting a tolerant and non-inflammatory environment [121].

In addition to the PD-L1/PD-1 interaction, the PD-L1/CD80 interaction is emerging as a significant pathway in terms of regulating autoimmunity. Sugiura et al. created an antibody that binds to CD80, disrupting the PD-L1/CD80 interaction while mostly sparing CD80’s interaction with both CD28 and CTLA-4. Administration of the novel CD80 antibody to mice with model autoimmune conditions, including rheumatoid arthritis and autoimmune encephalomyelitis, showed significant improvement in clinical and histologic scores of disease severity. Additionally, administration of the PD-L1/CD80 interaction blocking antibody decreased inflammatory markers such as IL-17 and IFN-γ as well as significantly decreased lymphocyte infiltration into glandular tissue in the case of mouse-modeled Sjogren syndrome [122].

The PD-L1/PD-1 interaction is crucial in the regulation of lymphocyte responses in various other autoimmune diseases. PD-L1’s absence has been shown to exacerbate a variety of immune-mediated disease processes such as encephalomyelitis, lupus-like nephritis, and autoimmune dilated cardiomyopathy in mouse models [123,124,125]. This shows that there is potential for using PD-L1 overexpression to decrease immune cell function and subsequent pathologies that may be associated with heightened immune cell function following organ transplant and in autoimmune conditions. 

### 4.3. Solid Organ Tolerance

In addition to PD-L1’s role in GVHD prevention, the role this ligand holds in organ transplantation has also been explored. Tanaka et al. studied the effects of PD-L1’s presence in mice that received a mismatched cardiac allograft that underwent concurrent immunosuppression with CTLA4Ig. Their study showed that blocking PD-L1 resulted in significantly reduced numbers of Tregs and significantly increased portions of alloreactive cytotoxic and IFN-γ-producing T cells within the spleen of allograft recipient mice. Regarding allograft function, Tanaka et al. demonstrated that PD-L1 blocking antibodies induced graft rejection within this same population of transplanted mice (Figure 2a,b). Moreover, PD-L1-deficient mice receiving heart transplants also had a significant reduction in graft survival. Histologically, it seems a possible cause of these effects in PD-L1 knockout mice was the increased vasculopathy within the cardiac tissue of mice [126].

Another study that focused on liver transplants observed how knockout of PD-1 receptors on murine T cells results in rapid liver allograft rejection in comparison to wild-type mice. Similarly, when treated with anti-PD-L1 antibodies, allograft recipient mice had significantly reduced survival rates as compared to allograft recipients not receiving anti-PD-L1 antibodies. Upon closer examination, allograft recipients who were treated with anti-PD-L1 antibodies revealed histologic liver sections that had significant parenchymal necrosis and vasculature inflammation, as well as drastic elevations in inflammatory markers like perforin/granzyme and iNOS mRNA within the grafted tissue [127].

An observational study of pediatric liver transplant patients observed an elevated ratio of PD-L1 to CD86 (a costimulatory molecule similar in function to CD80) within a population of DCs. There was a significant association between these patients’ diminished need for immunosuppressive therapy following transplant. In this same subset of patients, a relatively higher expression of PD-L1 on DCs was also directly correlated with higher levels of Tregs. However, it should be noted that this study did not show a correlation between a low PD-L1:CD86 ratio and low levels of Tregs [128]. Together, these studies reveal that PD-L1 may serve to protect allograft function by preventing immune-mediated tissue damage and promoting graft survival. However, further studies are needed to definitively prove or dispute PD-L1’s role in solid organ transplantation.

**Figure 2 cells-12-01609-f002:**
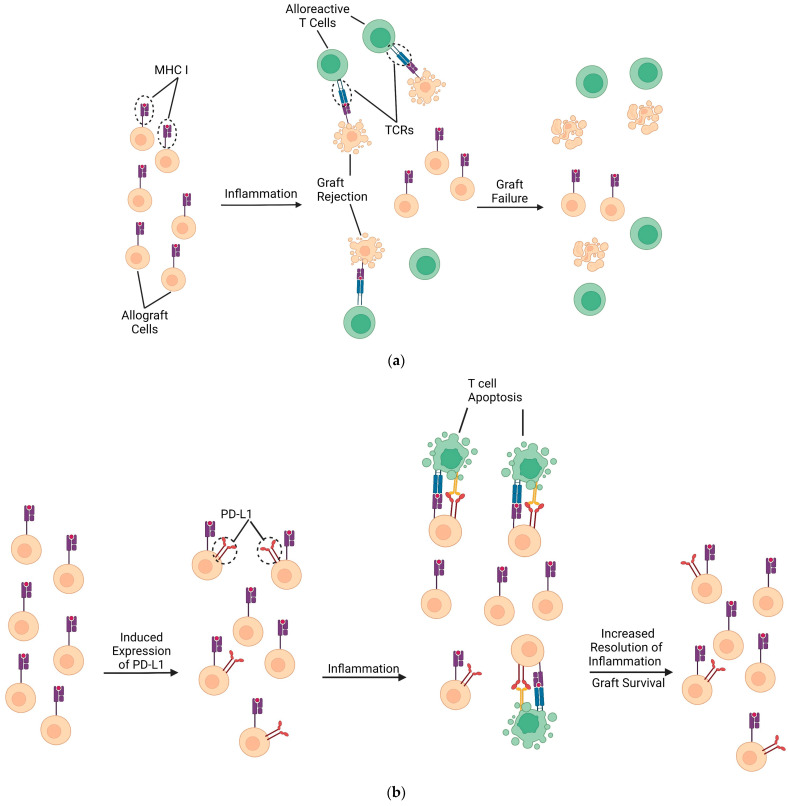
Proposed protective effects of PD-L1 during transplantation. (**a**) Limited PD-L1 expression by transplanted cells provides less protection from cytotoxic T cell infiltration (and less protection from helper T cell infiltration). Inflammation progresses to acute rejection and eventual graft failure as the tissue loses viable graft cells. (**b**) PD-L1 may be upregulated by methods such as IFN-γ administration or hydrodynamic gene transfer onto transplanted DCs and certain graft tissues such as hepatocytes [93,116,118,129]. If PD-L1 upregulation is sufficient to overcome CD80 sequestration/signaling, a reduction in alloreactive T cells would prevent graft cell induced apoptosis. Ideally, this technique would preserve graft function by preventing T cell infiltration.

### 4.4. PD-L1 Viral Reactivation

The potential of PD-L1 to prevent transplant rejection and graft survival must be weighed against its immunosuppressive properties, especially in the case of viral reactivation. The expression of PD-L1 on CD8+ T cells decreases the proliferative capacity and cytotoxic function of these cells. Given these effects, it is not surprising that many viruses induce PD-L1 overexpression, thereby decreasing the host cytotoxic T cell response and contributing to viral immune escape. Recently implicated viruses include Hepatitis B virus (HBV), HIV, Epstein–Barr virus, hantaviruses, and even coronaviruses [130,131,132,133,134]. Consequently, any therapy involving the upregulation of PD-L1 represents a clear pathway for the reactivation of underlying viral infections.

However, the effect of any immunotherapy interacting with PD-L1 is not so simple. While one would expect that any immunotherapy downregulating PD-L1 would lead to increased immune function and, therefore, decrease viral reactivations, there is clinical evidence of increased rates of HBV reactivation in cancer patients being treated with anti-PD-L1 monoclonal antibodies [135]. The exact mechanism by which this occurs is still unclear. Potential theories include the role of PD-1 in preventing immune-associated hepatocyte damage or the role of PD-1 in Treg apoptosis [43,136,137]. Overall, there is much to be discovered about the effects of PD-L1 immunotherapies in vivo.

Additionally, the potential adverse effects of PD-L1-induced immunosuppression must be viewed in the context of the current gold standard of post-transplant and GVHD immunosuppressive therapies, such as calcineurin inhibitors and corticosteroids. The high rates of infection in post-transplant patients due to immunosuppression are already well understood [4,138]. This is especially evident in the occurrence of viral reactivation. Traditional immunosuppressive therapies after solid organ transplant have been strongly implicated in the reactivation of Epstein–Barr virus, HBV, BK polyomavirus, and many more [138,139,140,141]. Overall, the potential immunosuppressive effects of PD-L1 must be further investigated, as it represents a promising alternative to current traditional immunosuppressive therapies.

## 5. Current State of PD-L1/PD-1 Targeting Therapies

Pharmaceutical targeting of PD-L1 has primarily focused on the inhibition of excess PD-L1 produced by cancer cells. Several monoclonal antibodies have been approved to target the PD-L1/PD-1 interaction, including pembrolizumab, nivolumab, and cemiplibab. Atezolizumab, avelumab, and durvalumab are all approved to target and inhibit PD-L1 [130]. Each of these therapeutics seeks to inhibit the immunosuppression seen in tumor cells that overexpress PD-L1, preventing immune tolerance in cancerous growth. However, these treatments have also been found to lead to the development of self-reactive T cells and have been linked to the development of autoimmune and inflammatory diseases [131].

In the absence of PD-L1/PD-1-induced anergy, patients receiving treatment that inhibits this pathway are at increased risk for developing inflammatory arthritis as well as hematologic diseases, including immune thrombocytopenia and autoimmune hemolytic anemia [132,133]. To circumvent some of the adverse effects noted in patients undergoing treatment with monoclonal antibodies, some researchers are pursuing combination therapies with small molecules such as osimertinib, an EGFR inhibitor, that mutually function to limit the over-activation of PD-L1. These combination therapies are in development for the treatment of lymphomas, NSCLC, colon cancer, and melanoma [134].

Other researchers have considered targeting non-membranous PD-L1 expression with small molecules that inhibit the expression of PD-L1 in its nuclear, cytoplasmic, or extra-cellular vesicle expression to ultimately reduce signaling sequelae [135,136]. Another avenue for investigation into PD-L1/PD-1 inhibition is more indirect by targeting its expression via known regulators of the PD-L1/PD-1 pathway, such as epigenetic modification and transcriptional and post-transcriptional regulators [137]. These emerging immunotherapies offer a variety of methods to enhance the body’s ability to target and kill cancer cells in a more precise manner. Researchers’ focus on PD-L1 and its role in the protection of cancer through anergy has resulted in prolonged lives for many. However, up until this point in time, researchers’ goal with PD-L1 targeting therapy has been to increase immune response via inhibition of the ligand. Therapeutic induction of PD-L1 expression seeks the opposite goal from most current pharmacologic agents in development—to promote local immunosuppression.

## 6. Conclusions

One major limitation shared by hematopoietic cellular therapies and solid organ transplantation is the occurrence of graft-versus-host disease (GVHD) and graft failure, respectively. In both cases, an overactive immune system targets and destroys functional tissue, similar to autoimmune diseases.

Promising targets for future immunotherapy include the interaction of PD-L1 with PD-1 or CD-80, which play a role in regulating T cells. Studies have shown that overexpression of PD-L1 in GVHD models improves survival and reduces pro-inflammatory cytokine secretion. Similarly, stimulating PD-L1 during organ transplantation prolongs the graft’s lifespan and reduces rejection rates. Additionally, in autoimmune disease models, increased PD-L1 expression leads to slower disease progression and limited tissue damage.

Although PD-L1 has primarily been studied in the context of cancer treatment, it has the potential to be a valuable tool in the management of allo- and auto-immunity. While most research has focused on blocking PD-L1 to inhibit immunosuppressive processes, enhancing PD-L1 activity could be a promising therapeutic approach as it dampens a destructive immune reaction. Further investigation is needed to understand how increased PD-L1 expression and availability in the context of GVHD, autoimmunity, and organ transplantation can promote immune tolerance.

## Figures and Tables

**Figure 1 cells-12-01609-f001:**
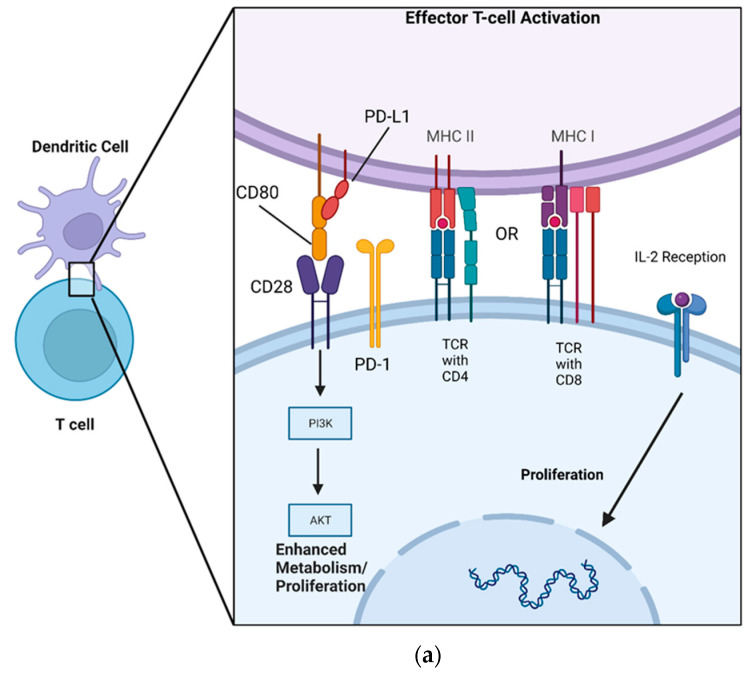
A DC acts as an APC for both CD8+ and CD4+ T cells via MHC I and MHCII, respectively, allowing for co-stimulatory/inhibitory signaling to occur. (**a**) MHC presentation of a complimentary peptide to the TCR allows for CD80’s activation of CD28 on the T cell, which subsequently enhances T cell metabolism and proliferation via the PI3K/AKT pathway. PD-L1 is sequestered by CD80, forming a heterodimer. CD80 simultaneously blocks the activation of PD-1 while activating CD28. (**b**) MHC presentation of a complimentary peptide to the TCR, followed by PD-L1 recognition by PD-1, induces phosphorylation of ITSM on the intracellular portion of PD-1. Phosphorylated ITSM then interacts with SHP-2, inhibiting the PIK3/AKT signaling cascade and limiting T cell production of IL-2.

## Data Availability

Not applicable.

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
