# Peer review of "PD-L1’s Role in Preventing Alloreactive T Cell Responses Following Hematopoietic and Organ Transplant"

_cells, 2023, doi:10.3390/cells12121609_

Round 1

Reviewer 1 Report

The writers present an elegant review about a highly relevant topic in current transplantation science, which is the role of immune checkpoint activation via PD-L1 on gvhd and graft rejection after HCT or organ transplantation. The review both provides a clear in-depth explanation of the mechanisms involved, as well as clinical potential of exploiting immune checkpoints in gvhd or graft rejection. Some suggestions are made to improve the review for publication.

- line 45: the writers state that “alloreactive T cells are generated”: might be more suitable to change generated into activated, please adjust.

- line 64: stating that GvHD is an autoimmune condition is not accurate, please adjust.

- line 68-71: not true that skin symptoms are early manifestations of aGvHD and liver/gut show at later stages. In aGvHD, one or more organs can be involved, ranging from skin, liver, gut, but also lungs. When only skin is involved: this is generally stage I aGvHD, and when other organs are (also) involved this makes aGvHD stage II-IV depending on severity of symptoms. There is also aGvHD of only gut and lungs, without skin involvement. So skin is not just an early manifestation of aGvHD but indicates a different stage. Please correct and rewrite this.

- line 126: this paragraph needs a bit more explanation, since the current statement is a bit blunt. Would be helpful to summarize here in one sentence that aGvHD and cGvHD have a similar mechanism in initiation of disease in alloreactive T cell activation, despite their differences in symptoms. Therefore, an interruption of alloreactive T cell activation…etc.

- to chapter 2: information about role of T cells in graft rejection after HCT is missing. Please consider adding this to the review.

- please consider discussing the potential effect of checkpoint induction on viral reactivations after HCT

- the term HSCT is not suitable in your review since HSCT grafts are stem cell only and do not contain T cells, and you are discussing allogeneic T cell reactions from donor T cells that are present in the graft; change into hematopoietic cell transplantation (HCT), as HCT grafts do contain both stem and immune cells.

Minor:

- please find consensus in choosing either chronic GvHD, cGvHD, or cGVHD; also for aGvHD or aGVHD.

- add chapter number to all paragraphs or to none of the paragraphs (currently only 2.1 is numbered)

Author Response

We are grateful to this reviewer for his insightful comments. "The writers present an elegant review about a highly relevant topic in current transplantation science, which is the role of immune checkpoint activation via PD-L1 on gvhd and graft rejection after HCT or organ transplantation. The review both provides a clear in-depth explanation of the mechanisms involved, as well as clinical potential of exploiting immune checkpoints in gvhd or graft rejection..."  Here are our responses point-by-point:

Question 1: - line 45: the writers state that “alloreactive T cells are generated”: might be more suitable to change generated into activated, please adjust.

We agree and have changed the sentence appropriately.

Question 2: line 64: stating that GvHD is an autoimmune condition is not accurate, please adjust.

Answer: We agree and have changed this language to reflect our agreement.

Question 3: - line 68-71: not true that skin symptoms are early manifestations of aGvHD and liver/gut show at later stages. In aGvHD, one or more organs can be involved, ranging from skin, liver, gut, but also lungs. When only skin is involved: this is generally stage I aGvHD, and when other organs are (also) involved this makes aGvHD stage II-IV depending on severity of symptoms. There is also aGvHD of only gut and lungs, without skin involvement. So skin is not just an early manifestation of aGvHD but indicates a different stage. Please correct and rewrite this.

Answer:We agree and have updated the sentence accordingly, line 68-74.

Question 4: - line 126: this paragraph needs a bit more explanation, since the current statement is a bit blunt. Would be helpful to summarize here in one sentence that aGvHD and cGvHD have a similar mechanism in initiation of disease in alloreactive T cell activation, despite their differences in symptoms. Therefore, an interruption of alloreactive T cell activation…etc.

Answer:We agree and have updated the sentence accordingly, line 121-132.

Question 5: - to chapter 2: information about role of T cells in graft rejection after HCT is missing. Please consider adding this to the review. 

-please consider discussing the potential effect of checkpoint induction on viral reactivations after HCT

- the term HSCT is not suitable in your review since HSCT grafts are stem cell only and do not contain T cells, and you are discussing allogeneic T cell reactions from donor T cells that are present in the graft; change into hematopoietic cell transplantation (HCT), as HCT grafts do contain both stem and immune cells.

Answer: We agree with it and have updated the information to reflect it, line 133-162.

Minor:

Question 6: - please find consensus in choosing either chronic GvHD, cGvHD, or cGVHD; also for aGvHD or aGVHD.

-add chapter number to all paragraphs or to none of the paragraphs (currently only 2.1 is numbered)

Answer: We agree and have made the necessary corrections.

Reviewer 2 Report

The review from Handelsman an colleagues "PD-L1’s Role in Preventing Alloreactive T Cell Responses Fol-lowing Hematopoietic and Organ Transplant" is well written and very informative.

There are only some minor suggestions.

Fig. 2 is very complicated, maybe the authors could simplify the figure for better and easy understanding.

Under "5. Current State of PD-L1/PD-1 Targeting Therapies" the authors bring up a very important point, however, this section could be a little more detailed and more linked to the topic of the review.

The conclusion could be more informative and addressing the importance of this topic.

Author Response

We thank the reviewer and consider the comments to be highly intriguing. " The review from Handelsman and colleagues "PD-L1’s Role in Preventing Alloreactive T Cell Responses Following Hematopoietic and Organ Transplant" is well written and very informative."... We have point-by-point responses all comments:

Question 1: Fig. 2 is very complicated, maybe the authors could simplify the figure for better and easy understanding.

Answer: We agree with the reviewer and have therefore changed Figure 2.

Question 2: Under "5. Current State of PD-L1/PD-1 Targeting Therapies" the authors bring up a very important point, however, this section could be a little more detailed and more linked to the topic of the review.

Answer: We updated the content in this section in order to agree with the reviewer.

Question 3: The conclusion could be more informative and addressing the importance of this topic.

Answer: We agreed this point and have revised our conclusion, line 505-522.